# OpenReview forum: "Breaking Bad: Exploring the Dangers of LLM-generated Misinformation from Fringe Social Media"
_NeurIPS.cc/2025/Workshop/Reliable_ML — NeurIPS 2025 - Reliable ML Workshop_

### Official Review · Reviewer_cJex · 2025-09-18
**Timely research on misinformation in short-form social posts generated by unmoderated LLM**

**Rating:** 7
**Confidence:** 3

**Review:**

## Summary

This paper investigates the risks posed by unmoderated LLM services hosted on fringe social networks and evaluates existing detection methods on misinformation in short-form social media posts generated by these services.

Specifically, two prompting strategies, topic-based and evidence-based, are adopted on two AI characters from Gab AI, Covid 19 Skeptic and Charlie Conspiracy. Upon verification by experts, 90.4% of the 460 generated responses constitute misinformation, which is significantly higher than moderated systems such as ChatGPT, which refuses to perform the task.

Two misinformation detection methods, zero-shot and few-shot prompting with and without CoT, are evaluated. While zero-shot detection performs poorly, few-shot detection is substantially better, particularly when combined with CoT, with Llama-3 achieving 93.03% overall accuracy.

General-purpose AI-content detectors, TweepFake and Fast-DetectGPT, are also evaluated. While TweepFake achieves 62.5% overall accuracy, the performance is not ready for real-world deployment.

Overall, this paper points out the challenge in misinformation from unmoderated LLM-generated and short-form social media posts. From the evaluation, few-shot detection prompting with CoT examples can be a potential solution.

## Strengths

1. **Timely research**. As LLMs are more and more used on social media, this study on LLM-generated misinformation is timely and practical. Different from mainstream research, this paper suggests unique challenges posed by unmoderated LLMs and short-form responses in the form of social media posts.
2. **Well-conducted empirical evaluation**. The evaluation of misinformation generation and detection involves human experts and multiple mainstream LLMs and tools, making it convincing and providing insightful results.
3. **Clarity in writing**. This paper is well-structured and easy to follow, with adequate background, examples, and explanation.
4. **Relevance to the workshop topic**. LLM-generated misinformation can be viewed as adversarial or low-credibility data, which aligns with the topic of the workshop.

## Weaknesses

1. **Limited scope**. The LLM is limited to one platform (Gab AI) and one topic (COVID-19), leaving it unclear whether the conclusions hold in a more general case.
2. **Does not involve negative examples**. The evaluation does not include factually true claims about COVID-19, which are considered as negative examples for detectors. It is unclear how the evaluated methods perform on these true claims.
3. **Lack of sensitivity analysis**. The performance of few-shot detection can depend on the selection of examples. This paper does not analyze the sensitivity to such selection.
4. **Lack of failure case analysis**. Cases where detectors fail are not examined to identify potential failure modes.

## Suggestions for Authors

1. Expand the scope of evaluation, both on more unmodified LLMs and on more topics such as climate and political topics.
2. Include true claims, either human-written or LLM-generated, to complement the evaluation.
3. Perform sensitivity analysis, failure case analysis, and more ablation studies. Apart from the above-mentioned experiments, it is also worthwhile to study how the length of posts affects detection performance to show the unique challenges in short-form posts.

## Ethics

Demonstrations of how unmoderated LLMs can generate misinformation and publishing the prompting methods used can risk misuse if replicated. Also, the use of real journal/organization/drug names can risk their reputation. One possible mitigation is to replace critical information with placeholders.

---

### Official Review · Reviewer_t5Fn · 2025-09-24
**Review response**

**Rating:** 6
**Confidence:** 3

**Review:**

The paper audits unmoderated LLMs embedded in a fringe platform (Gab AI) and shows they can mass-produce short COVID-19 misinformation with minimal effort. The study highlights the risk of short-form, platform-integrated LLM misinformation and a practical mitigation (few-shot detectors).

## Strengths
- The paper addresses a timely and important problem by focusing on unmoderated, in-platform LLMs on fringe social networks.
- The experiments provide a clear and compelling contrast by evaluating zero-shot detection versus few-shot detection with chain-of-thought on the same generated posts
- Paper includes human expert evaluation

## Weakness / Suggestions
- The experiment was done on single topic (COVID-19) and one platform (Gab)
- Although there are only 2 medical experts participated in annortation, would be great to see inter rater reliability or some metric to show how credible and consistent the answer was